# Atomic faulting drives exceptional toughness in low thermal expansion chromium alloys

Chengyi Yu [1,8], Honghui Wu [2,8], Huihui Zhu[2,8], Xin Chen [1], Qinghua Zhang [3], Yujie Chen[4], Lin Gu [4], Qiang Zhang [5], Matthias D. Frontzek [5], Yan Chen[5], Ke An [5], Lunhua He [3], Kenichi Kato[6], Shogo Kawaguchi [7], Zeyu Qiao [1], Meisa Zhou[2], Yili Cao[1], Qiang Li[1], Jinxia Deng[1], Kun Lin [1] & Xianran Xing [1] ✉

Endowing functional properties with mechanical responses in traditional metals has been a frontier topic, akin to transforming base metal into gold. Chromium and its alloys, with their functional deficiencies and limited ductility, serve as typical examples. Herein, we report a $Cr_{96}Fe_4Ge_{1.3}B_1$ alloy that unifies low thermal expansion (LTE, $\alpha_l = 1.79 \times 10^{-6} K^{-1}$, 200 − 315 K) with exceptional toughness (240.2 J·cm$^{-3}$). The enhancement in mechanical responses is primarily attributed to layered $Cr_2B$ intermetallic precipitates, which ameliorate interfacial cohesion and simultaneously refine the grain structure. The weakened interlayer interactions within the Cr-B layers facilitate the nucleation and movement of numerous tiny stacking faults in precipitates, efficiently alleviating strain energy and resulting in marked work-hardening ability. Additionally, antiferromagnetic fluctuations in the BCC matrix contribute to the unique LTE behavior. This paves the way for the design of high-performance alloys featuring layered-symmetry precipitates.

Chromium, one of the three major black metals on Earth, is typically characterized by a stable cubic phase at ambient temperature[1]. Unlike widely used steels, which share the same crystal symmetry and slip systems, chromium exhibits inadequate ductility and poor workability[2–4]. These deficiencies primarily arise from the outer electronic configuration ($3d^54s^1$), resulting in stronger metallic bonding that renders dislocation motion harder[5,6]. Additionally, the inherent brittleness of the grain boundary further exacerbates the degradation of its mechanical properties[7,8]. Consequently, the potential for chromium-based alloys as

functional or structural materials remains in its infancy, but is urgently needed[9,10].

The close coupling between the spin, orbit, and lattice in pure chromium ($T_N = 312$ K) suggests that exploring zero or low thermal expansion (ZTE/LTE) in chromium-based alloys could be a promising avenue[11–16]. Such dimensional invariance is pivotal for high-precision instruments, exemplified by the ultra-stable components required for gravitational wave detection and space telescopes, where even microscopic thermal fluctuations can compromise performance[17–20]. However, emerging technologies increasingly demand materials that combine this

[1]Institute of Solid State Chemistry, School of Chemistry and Biological Engineering, University of Science and Technology Beijing, Beijing, China. [2]Beijing Advanced Innovation Center for Materials Genome Engineering, State Key Laboratory for Advanced Metals and Materials, Institute for Carbon Neutrality, University of Science and Technology Beijing, Beijing, China. [3]Institution of Physics, Chinese Academic of Science, No.8, 3rd South Street, Zhongguancun Haidian District, Beijing, China. [4]Beijing National Center for Electron Microscopy and Laboratory of Advanced Materials, School of Materials Science and Engineering, Tsinghua University, Beijing, China. [5]Neutron Scattering Division, Oak Ridge National Laboratory, Oak Ridge, TN, USA. [6]RIKEN SPring-8 Center, 1-1-1 Kouto, Sayo-cho, Hyogo, Japan. [7]Japan Synchrotron Radiation Research Institute (JASRI), Sayo-cho, Hyogo, Japan. [8]These authors contributed equally: Chengyi Yu, Honghui Wu, Huihui Zhu. ✉e-mail: xing@ustb.edu.cn

thermal stability with robust mechanical integrity and environmental resilience—a synergy rarely found in conventional ZTE alloys. Specifically, in applications like semiconductor lithography or cryogenic engineering, components must maintain atomic-level precision while withstanding complex mechanical loads and plasma/acid corrosion. Consequently, the inherent corrosion resistance of chromium alloys, when engineered for high toughness, provides a unique solution for these extreme environments. To achieve this, both magnetic and non-magnetic atoms are incorporated into the chromium sublattice to modulate electron-spin interactions, emerging in a single-phase ZTE alloy with a composition of $Cr_{0.96}Fe_{0.04}Ge_{0.13}$ covering a temperature range of 200–315 K ($\alpha_l = 1.08 \times 10^{-6} K^{-1}$). The focus now shifts towards optimizing the mechanical response for practical applications[21-25].

Constructing soft/hard heterostructures is an effective way to achieve ZTE alloys with superior mechanical properties, as exemplified by the α-Fe/$Ho_2Fe_{17}$ dual-phase alloy[26]. In comparison, complex intermetallic compounds are typically avoided in structural materials for their tendency to induce material brittleness[27-30]. However, we propose that the layered $Cr_2B$ precipitate (Space group: $Fddd$) imparts high toughness to chromium-based ZTE alloy[31-35], improving it by an order of magnitude. Such a high mechanical response is ascribed to the interface strengthening and the movement of atomic-scale faults during the loading of the $Cr_2B$ precipitates. The electronic density of states clarifies that these stacking faults originated from weaker interlayer interactions within the Cr-B layers than the Cr-Cr layers. Our study provides a pathway toward developing chromium-based alloys and supports their functional and structural applications[36,37].

## Results and discussion

The chromium-based alloys with the composition $Cr_{96}Fe_4Ge_x$ ($x = 0$, 0.3, 1.3, and 2.5, $at.\%$, labeled as S-1, S-2, S-3 (CFGB$_0$), and S-4) were designed through chemical tailoring (Supplementary Fig. 1 and Supplementary Discussion 1). The body-centered-cubic symmetry (BCC, $Im$-$3m$, Supplementary Fig. 2) was unveiled by synchrotron X-ray diffraction (SXRD). Intriguingly, the dilatometer thermal expansion shifts from negative thermal expansion (NTE) in S-1 ($\alpha_l = -11.49 \times 10^{-6} K^{-1}$, 200–260 K) to positive thermal expansion (PTE) in S-4 ($\alpha_l = 3.41 \times 10^{-6} K^{-1}$, 200–335 K), as shown in Fig. 1a. A favorable zero thermal expansion (ZTE) was achieved in S-3 (CFGB$_0$) alloy covering room temperature ($\alpha_l = 1.08 \times 10^{-6} K^{-1}$, 200–315 K). This result is further verified by the evolution of lattice parameters obtained from synchrotron X-ray diffraction (SXRD) and neutron powder diffraction (NPD) (Fig. 1b and Supplementary Fig. 3). Macroscopic magnetic measurements demonstrate an antiferromagnetic-paramagnetic transition ($T_N = 315$ K) in S-3 (Supplementary Fig. 3). This transition, consistent with the thermal expansion behavior, indicates that the anomalous thermal expansion may stem from magnetic contributions. The atomic magnetic moments ($M$(Cr/Fe)) are arranged in layers in antiparallel order along the c-axis, exhibiting antiferromagnetism (Fig. 1c). Driven by the antiferromagnetic order to resist the normal lattice expansion mostly arising from phonons, lattice contraction is generated with the amplitude progressively increasing upon cooling, driving the ZTE behavior (Supplementary Discussion 2 and Supplementary Fig. 4)[14].

In the S-3 (CFGB$_0$) alloy, a low ultimate strength ($\delta_{US}$) of 0.61 ± 0.04 GPa, and compressibility ($\varepsilon_f = 9.03 \pm 0.8\%$) until failure

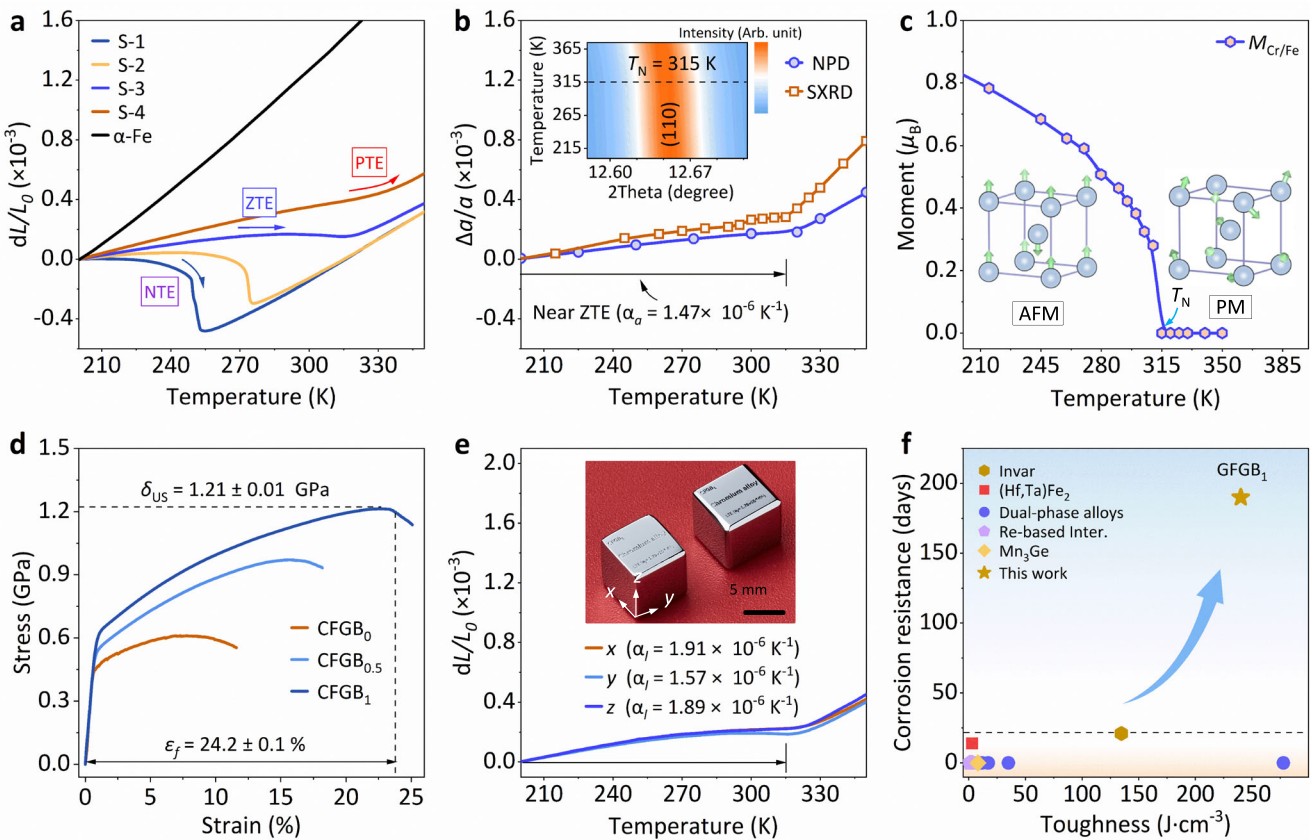

**Fig. 1 | Thermal expansion and mechanical properties of the alloy series.**
**a** Thermal expansion behavior of $Cr_{96}Fe_4Ge_x$ ($x = 0$, 0.3, 1.3 and 2.5, $at.\%$, labeled as S-1, S-2, S-3 (CFGB$_0$) and S-4) as measured by a dilatometer. **b** Lattice thermal expansion performance of the S-3 alloy as determined by SXRD and NPD, insert is the SXRD profiles of the (110) reflection. **c** Magnetic moment ($M_{Cr/Fe}$) and magnetic structure of the S-3 alloy determined by NPD. **d** Engineering compressive stress-

strain curves for the ($Cr_{96}Fe_4Ge_{1.3}B_x$, $x = 0$, 0.5 and 1, labeled as CFGB$_0$, CFGB$_{0.5}$ and CFGB$_1$) alloys. **e** Thermal expansion performance of the CFGB$_1$ alloy along the different directions, respectively. Inserted the machined ultra-stable components. **f** Corrosion resistance versus toughness of classic ZTE/LTE alloys[57], such as Rare-earth based intermetallics (Re-based Inter.)[58].

were observed, limiting its potential usages (Fig. 1d). To enhance the mechanical responses, boron atoms ($Cr_{96}Fe_4Ge_{1.3}B_x$, $x = 0.5$ and 1, labeled as $CFGB_{0.5}$ and $CFGB_1$) were added into the alloy. It was found that both the ultimate strength ($\delta_{US} = 1.2 \pm 0.05$ GPa) and the compressibility at failure ($\varepsilon_f = 24.4 \pm 2.4$ %) are significantly improved with increased boron content in $CFGB_1$ alloy (Fig. 1d and Supplementary Fig. 5). Notably, the $CFGB_1$ alloy maintains low thermal expansion ($\bar{\alpha}_l = 1.79 \times 10^{-6} K^{-1}$, 200–315 K, Fig. 1e, Supplementary Fig. 6), even after a slight overall increase (Supplementary Discussion 3 and Supplementary Fig. 7). Additionally, it displays superior corrosion resistance over 190 days in a 3.5 $wt.\%$ NaCl solution, adopted to standardize the

dominant chloride-induced corrosion mechanism combined with high toughness 240.2 J·cm$^{-3}$ (Fig. 1f, Supplementary Fig. 8, Supplementary Table 1 and Supplementary Discussion 4). Thus expanding its potential uses in marine environments as ultra-stable components (Inserted in Fig. 1e)[38,39]. In addition, due to the natural heterostructure, the alloy exhibits superior thermal cycling stability in both thermal expansion and mechanical properties (Supplementary Fig. 9).

A dual-phase structure comprising the $Cr_2B$ precipitate (Space group: *Fddd*) within the BCC matrix (Space group: *Im-3m*) is manifested by SXRD and electro-probe microanalysis (EPMA, Fig. 2a, b). The $Cr_2B$ phase forms due to the enrichment of boron atoms at the grain

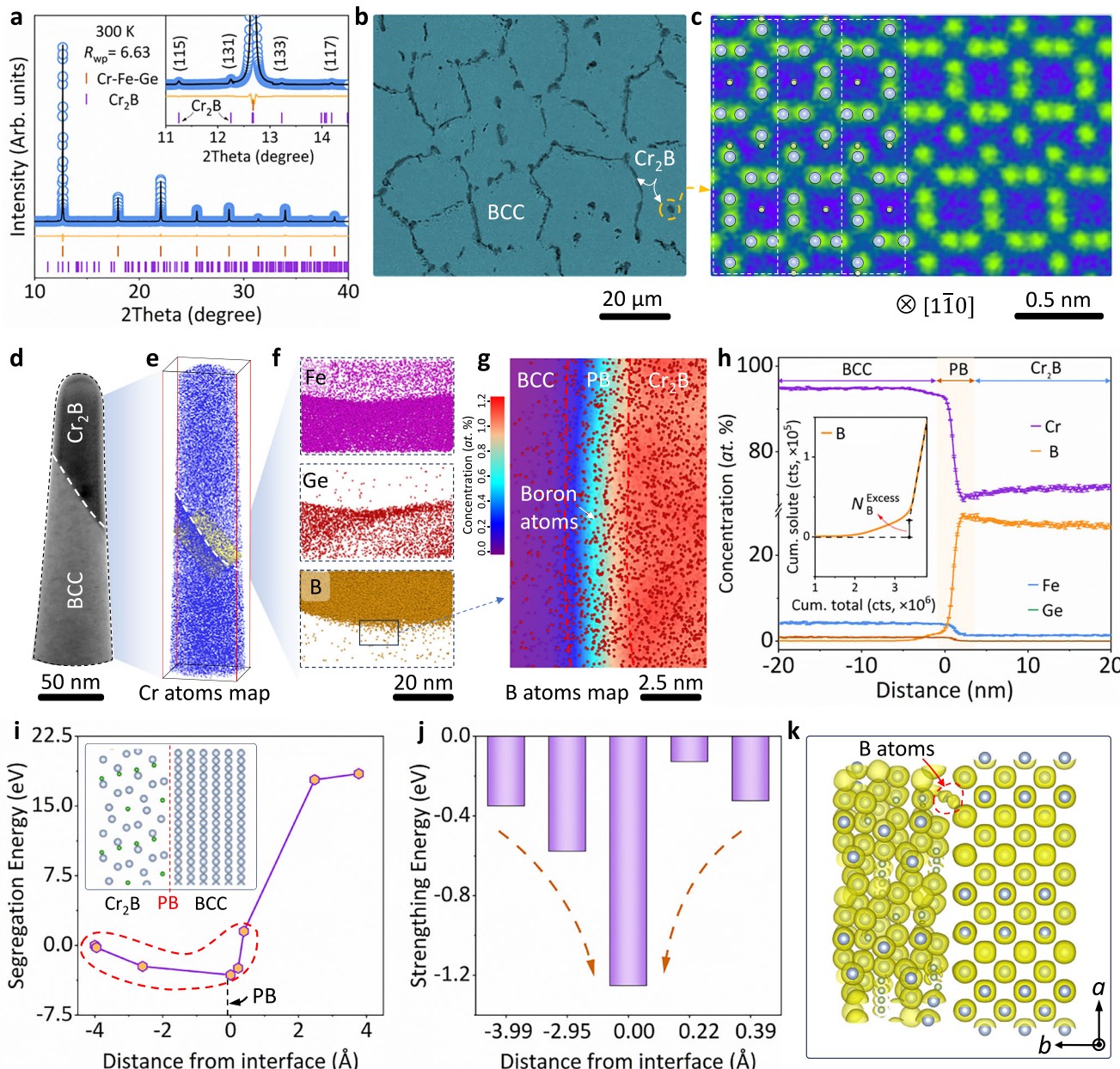

**Fig. 2 | Microstructure of the dual-phase alloys. a** Rietveld refinement of the SXRD profile for the $CFGB_1$ alloy. **b** EPMA image of the $CFGB_1$ alloy, showing the BCC matrix (gray grains) and $Cr_2B$ precipitates (black grains). **c** HAADF-STEM of the $Cr_2B$ phase along the [$1\bar{1}0$] zone axis. **d** 3D APT results determined by TEM. **e** Cr atom maps reconstructed using 3D-APT, highlighting the nanoprobe phase interface with the BCC matrix (bottom) and the $Cr_2B$ phase (top). **f** Detailed view from the selected yellow box in (**e**), illustrating the distributions of Fe, Ge, and B atoms. **g** Two-dimensional concentration contour of boron atom distribution. **h** Proximity

histogram for the 4-$at.\%$ B iso-concentration surfaces by using the complete probe, showing the distribution of Cr, Fe, Ge, and B atoms. Inserted the calculation of Gibbs interfacial excess of boron. Error bars represent standard deviation. **i–j** Segregation energy and strengthening energy at the phase boundaries with boron atoms at various sites, including an inset of the crystal structure model. **k** Charge density distribution at the phase boundary when boron atoms occupy interstitial sites.

boundaries (4.68 ± 0.02 mass %, Supplementary Fig. 10). Especially, the presence of $Cr_2B$ precipitate significantly reduces the grain size of the BCC matrix from 190 μm ($CFGB_0$) to 25 μm ($CFGB_1$, Supplementary Fig. 11), which in turn enhances the yield strength $\sigma_y$ through the grain refinement strengthening effect (Supplementary Discussion 5 and Supplementary Fig. 12). Detailed atomic arrangements along the [1$\bar{1}$0] zone axis illustrates the hierarchical ordering of chromium atoms along the $c$-axis in the orthorhombic structure (Fig. 2c and Supplementary Fig. 13). Additionally, EPMA and atom probe tomography (APT) were combined to investigate compositional fluctuations at the phase interface (Fig. 2d–f)[40,41]. Boron atoms were detected at the phase interface, as shown in Fig. 2g, h. And the comparison of Gibbs interfacial excess values, $\Gamma_B = 4.17 \pm 0.26$ nm$^{-2}$ and $\Gamma_{Cr} = -0.93 \pm 0.26$ nm$^{-2}$, further confirms the enrichment of boron atoms at the interface (Supplementary Fig. 14 and Supplementary Discussion 6)[42,43], suggesting its potential contribution to interfacial strengthening[44–46].

To elucidate the pivotal role of boron atoms in strengthening phase boundaries, first-principles calculations based on density functional theory (DFT) were conducted. Structural models with boron atoms positioned at various crystal sites along the phase boundary (Supplementary Fig. 15) were initially constructed. It should be noted, however, that the measured composition could be influenced by field-dependent effects such as the loss of boron during APT analysis[47]. The segregation energy ($E_{seg.}$) exhibits negativity and is lower compared to other sites at the phase boundary (Fig. 2i), suggesting a tendency for boron atoms to enrich at the phase boundary. Importantly, a lower strengthening energy ($\Delta E_{SE}$, Fig. 2j) is displayed by the structural configuration of the boron atoms at the phase boundary, implying enhanced boundary strength. It is further illustrated by the charge density distribution at the phase boundary when boron atoms reside in the interstitial site (Fig. 2k). Strong electronic interaction with the metallic host atoms is expressed by the boron atoms, resulting in the cohesive strengthening of the boundary region.

The deformation behavior of the dual-phase alloy ($CFGB_1$) was investigated by in situ neutron diffraction experiments (Supplementary Fig. 16 and Supplementary Fig. 17)[48–50]. It was observed that both the BCC matrix and $Cr_2B$ precipitates underwent elastic deformation and uniformly withstood the load during Stage I (Fig. 3a). As the loading stress increases (~0.62 GPa, Supplementary Fig. 18), the matrix progressively yields, verifying the domain role in enhancing the yield strength ($\sigma_y$) by the grain refinement effect. Meanwhile, the additional stress is transferred to the precipitates (Fig. 3b), while the precipitates remain elastic (0.62 – 0.79 GPa, Stage II). During the plastic deformation stage, the slip system is activated, and dislocations begin to generate and multiply in the BCC matrix (Fig. 3c, Supplementary Fig. 19, and Supplementary Discussion 7), which dominates the deformation behavior in the alloy. However, the dislocations are hindered and pinned at the phase interface by the $Cr_2B$ precipitates, which do not provide such highly effective work-hardening capabilities (Fig. 3b and Supplementary Fig. 19). This may be attributed to the intrinsic strengthening of the $Cr_2B$ precipitates[35].

As a result, transmission electron microscopy (TEM) was employed to analyze microstructural evolution. In contrast to the BCC matrix, interestingly, the $Cr_2B$ precipitates exhibit a unique stacking fault slip mode when activated ($\varepsilon = 2.5\%$) under loading conditions (Fig. 3d–f). This stacking fault deformation originated from the interlayer slip of the {001} crystal plane (Supplementary Fig. 20 and Supplementary Discussion 8). With an increase in strain ($\varepsilon = 7.5\%$ and $\varepsilon = 15\%$), the stacking faults propagate and multiply (Fig. 3g–i), with interlayer spacings shrinking to nanometer scales. This demonstrates that $Cr_2B$ precipitates not only play a strengthening role but also actively participate in deformation, effectively transferring and releasing stress[51,52].

Stacking defects in the $Cr_2B$ phase are activated and slip on the non-metal (boron atoms) layers of the {001} crystal planes ($\varepsilon = 2.5\%$,

Supplementary Fig. 21) in the early stages. From a structural perspective, we decomposed the $Cr_2B$ phase into three distinct structural types along the $c$-axis (Supplementary Fig. 22): A (boron atomic layers), B, and C (chromium atomic layers). Notably, even with larger plastic deformations ($\varepsilon = 15\%$), stacking defects are generated only in the A-B layers (metal and non-metal layers), rather than the B-C layer (metal-metal layer, Fig. 4a, b). To elucidate the sliding mechanism of stacking faults in the $a$-$b$ plane, the generalized stacking fault energy (GSFE, Supplementary Fig. 23) was calculated. The stacking fault energy ($\gamma_{USF}$) along the [110] direction[53,54], which corresponds to the first local maximum point in the GSFE curve, is lower in the A-B layer compared to the B-C layer (Fig. 4c, d). This result further supports stacking fault deformation mode is {001} <110> within the metal and non-metal layers. Therefore, the boron atomic layer plays a key role in activating the atomic stacking faults in the $Cr_2B$ precipitates, releasing interfacial stress and improving the work-hardening ability of the alloy.

The chemical bonds and electronic structure of metal and non-metal layers (A-B and B-C layers) were further analyzed by the projected crystal orbital Hamilton populations (-pCOHP)[55,56]. Specifically, four Cr-B bonds within the A-B layers and eight Cr-Cr bonds in the B-C layers were found to dominate the interlayer fault slip behavior (Fig. 4e). The -pCOHP curves of the B2$p$-Cr3$d$ and Cr 3$d$-Cr3$d$ orbits illustrate strong bonding states below the Fermi level ($E_F$) and the spindle-shaped track overlap promotes stacking fault slip (Fig. 4f). A single Cr-B bond exhibits more strength than a Cr-Cr bond (Fig. 4g, Supplementary Table 2), however, the cumulative bond energies in the B-C metal layer surpass those in the A-B non-metal layer. As a result, the relatively weak interlayer bonding of the Cr-B layer is the driving force for the slip of atomic stacking faults.

In summary, we demonstrate the unification of low thermal expansion (LTE) behavior and mechanical responses in chromium-based alloys through precise chemical modulation. The anti-ferromagnetic ordering of the BCC matrix is identified as the key factor contributing to LTE. Enhanced toughness results from the strong chemical bonding of interstitial boron atoms at the local interface, coupled with the refined grain structure of the $Cr_2B$ precipitates. Furthermore, analysis of chemical bonds and electronic structure reveals that weakened metal–non-metal layer interactions promote atomic stacking fault formation, thereby ameliorating mechanical properties. This work offers valuable insights into strategies for advancing the applications of chromium and its alloys.

## Methods
### Materials and preparations
A series of Cr-Fe-Ge-B alloys was prepared by arc melting with elements of more than 99.9% under a high-purity argon atmosphere. The samples were turned over and melted four times to ensure homogeneity. Subsequently, annealing was conducted at 1423 K in an argon atmosphere for about 6 h, followed by quenching in liquid nitrogen ($LN_2$).

### Dilatometer thermal expansion and magnetization measurements
The dilatometer thermal expansion was conducted by an advanced thermo-dilatometer (NETZSCH DIL402). Magnetization measurements were taken with a Quantum Design physical property measurement system (PPMS) equipped with a liquid helium-cooled vibrating sample magnetometer (VSM).

### Crystal and magnetic structure
The synchrotron X-ray diffraction (SXRD) of the samples at room temperature was collected at the BL44B2 and BL02B2 beamlines ($\lambda = 0.45$ Å) in Spring-8, Japan. The magnetic structure was determined by neutron powder diffraction (NPD) and was conducted at the Neutron Scattering Division, Oak Ridge National Laboratory (ORNL), USA,

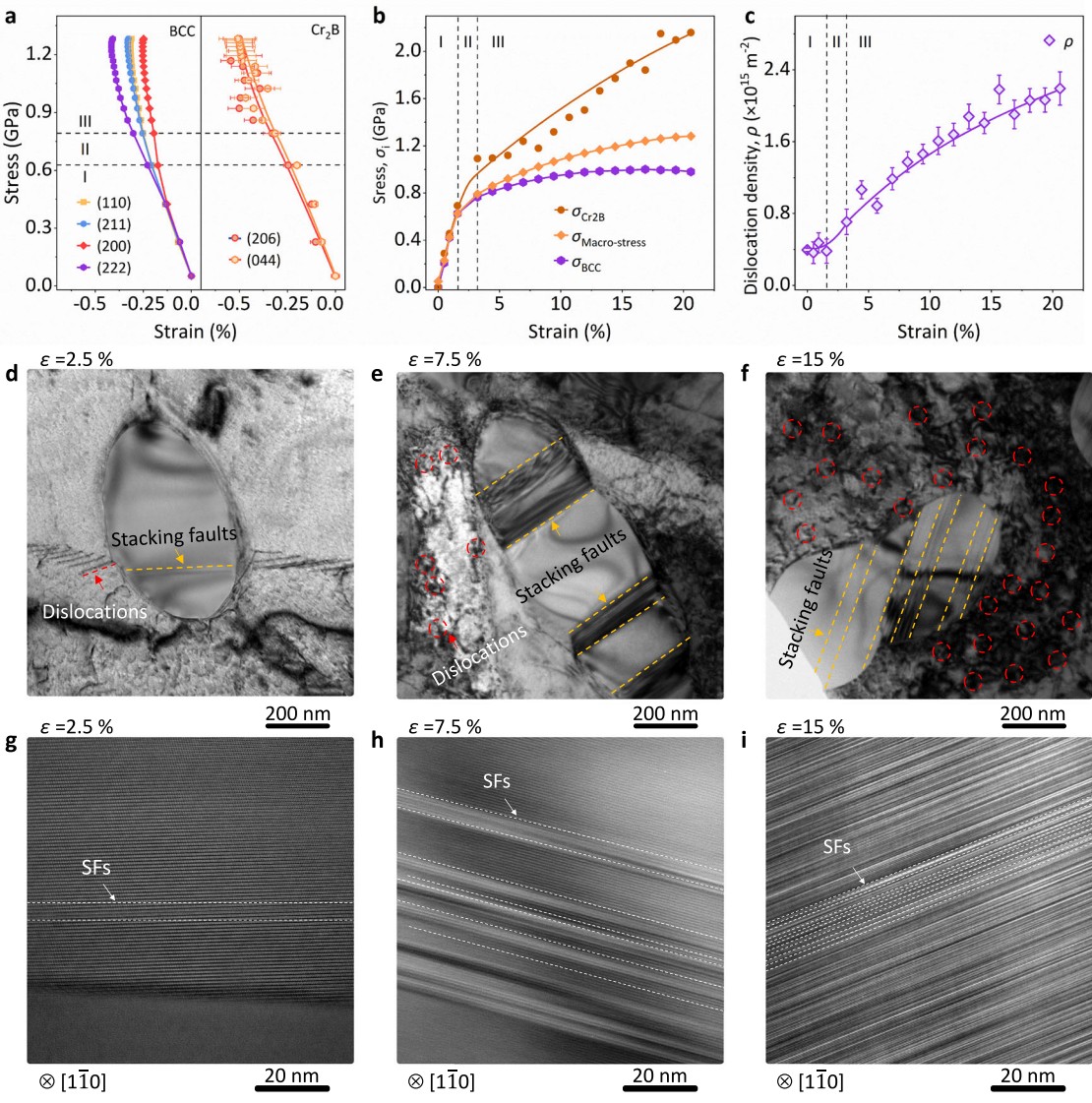

**Fig. 3 | Microstructure evolution of the alloy. a** Lattice strain of the CFGB$_1$ alloy along the loading direction (LD). Error bars represent standard deviation. The deformation zones correspond to elastic (I), load transfer (II), and plastic co-deformation (III) stages. **b** Specific stress partitioning in the BCC matrix and Cr$_2$B precipitates. Note that the strain is determined by in situ neutron diffraction without an extensometer. **c**, Dynamic dislocation densities ($\rho$) of the BCC matrix were determined by the modified *Williamson-Hall* (MWH) method. Error bars represent standard deviation. **d–f** Microstructure of the dual phase alloy at different strains $\varepsilon = 2.5$, 7.5, and 15 %, respectively. **g–i** The stacking faults in the Cr$_2$B precipitate along [1$\bar{1}$0] zone axis at different strain $\varepsilon = 2.5$, 7.5 % and 15%, respectively.

and a general-purpose powder diffractometer (GPPD) (90° bank) at the China Spallation Neutron Source (CSNS), Dongguan, China. In situ loading study by neutron diffraction was performed at the VULCAN beamline (BL-7) in Oak Ridge National Laboratory (ORNL), USA. The crystal structure and magnetic structure were analyzed using the FullProf Suite software.

## Mechanical properties

The stress-strain curves at room temperature were measured on a CMT4105 universal electronic compressive testing machine using a Φ 5 × 8 mm cylinder and an initial strain rate of 0.25 mm/min, performed three times for each sample. The toughness was calculated as the area under the compressive stress-strain curve.

## Microstructures and phase interface structure

Electron backscatter diffraction (EBSD) was performed using a TES-CAN MIRA 3 LMH SEM equipped with an Oxford Symmetry S2

detector. Data were collected at an accelerating voltage of 20 kV with a step size of 0.05–0.2 μm, and analyzed using HKL Channel 5 software (v.5.12). TEM specimens were prepared via FIB lift-out. High-resolution imaging was conducted on an FEI Titan G2 60–300 S/TEM operated at 300 kV. Image processing was performed using DigitalMicrograph (GMS 3.4). CAMECA Instruments LEAP 5000XR was used for the APT characterizations. The data were collected in voltage mode at a specimen temperature of 50 K, a pulse repetition rate of 200 kHz, and an ion collection rate of 0.5% per-field evaporation pulse. The APT data were reconstructed using AP Suite 6.3, and the reconstruction was calibrated using crystallographic elements retained in the data as represented by spatial distribution maps.

## First-principles calculations

The first-principles calculations were performed using density functional theory (DFT) within the Vienna ab initio simulation package (VASP). The segregation tendency of boron atoms at phase boundaries

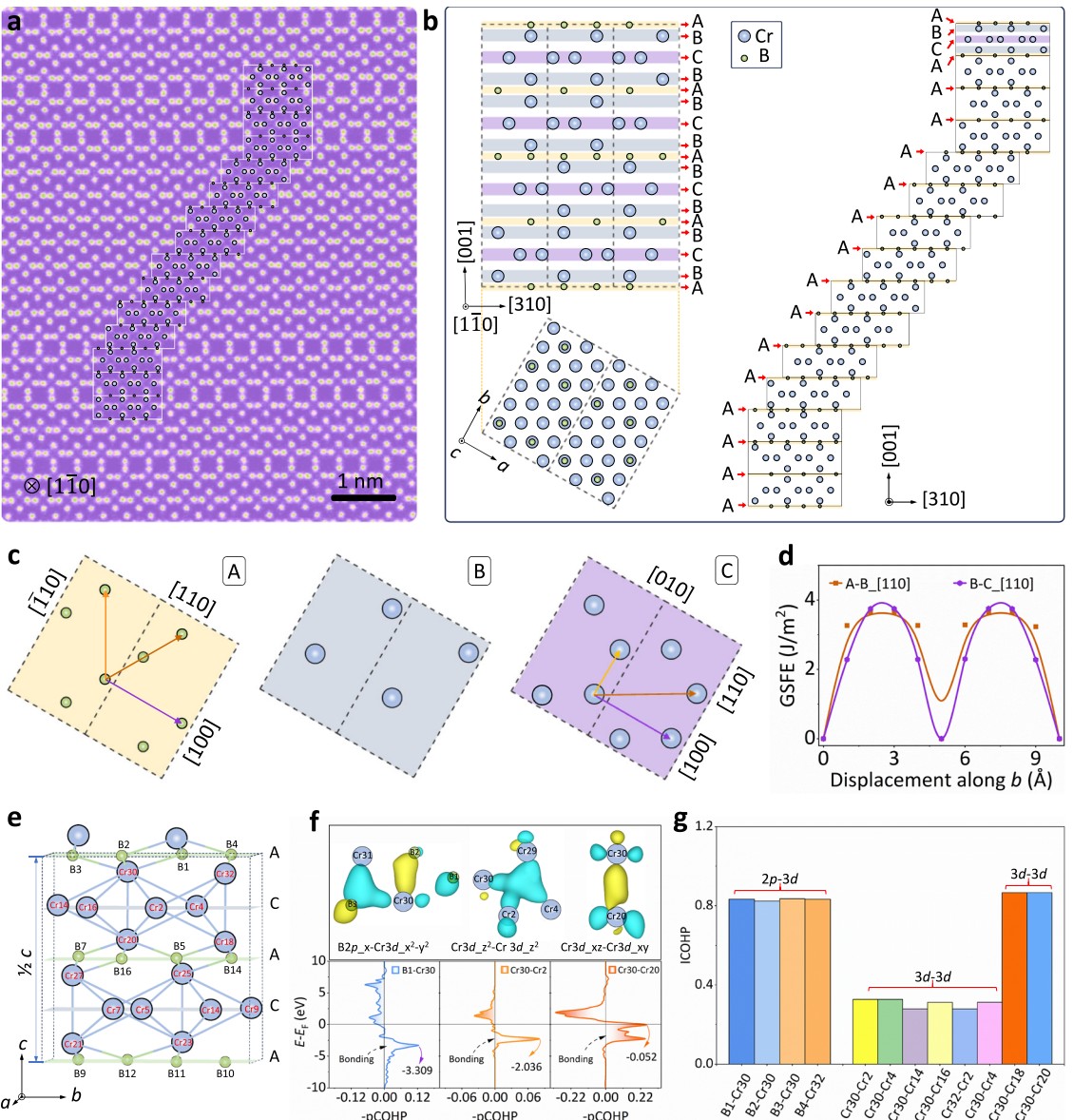

**Fig. 4 | Evolution of atomic stacking faults in the CFGB1 alloy. a** HAADF-STEM image of the Cr₂B phase along the [1Ī0] zone axis at $\varepsilon = 15\%$. **b** Detailed structure of the atomic stacking faults in the Cr₂B phase corresponding to (**a**). **c**, **d** Illustration of the crystal structure (A, B, and C layers) and the corresponding GSFE along the [110] direction, with different arrows indicating the slip directions. **e** The chemical bonding interactions between the layers (A-B and B-C layers). **f** The -pCOHP for the B 2$p$-Cr 3$d$ and Cr 3$d$-Cr 3$d$ orbitals. **g** The integration projected crystal orbital Hamilton population (ICOHP) of the B 2$p$-Cr 3$d$ and Cr 3$d$-Cr 3$d$ interactions in the A-B and B-C layers.

(PB) is determined by the segregation energy ($E_{seg}$) formula (1):

$$E_{seg} = (E_{PB}^B - E_{PB}) - (E_{Bulk}^B - E_{Bulk}) \qquad (1)$$

where $E_{PB}^B$ and $E_{PB}$ is the total energy of the PB model with and without boron atoms, respectively. Hence, a negative segregation energy signifies that a boron atom separating at the grain boundary is energetically advantageous. The strengthening energy was computed based on the Rice-Wang model formula (2):

$$\Delta E_{SE} = (E_{PB}^B - E_{PB}) - (E_{FS}^B - E_{FS}) \qquad (2)$$

where $E_{PB}^B$ and $E_{PB}$ represent the total energy of the simulated PB model with and without boron atoms, respectively. $E_{FS}^B$ and $E_{FS}$ denote the total energy of the cracked PB model with and without boron atoms, respectively. The generalized stacking fault energy (GSFE) is

calculated by shifting the respective crystal planes and adding an incremental slip in the various directions, followed by optimization with the ionic degrees of freedom fixed in the slip direction, which yields an energy $U(s)$. The stable stacking fault energy is also calculated with all ionic degrees of freedom relaxed. The GSFE $\gamma$-line is then obtained by formula (3):

$$\Psi_{(s)} = (U_{(s)} - U(0))/A \qquad (3)$$

where $A$ is the area of the slip plane in the supercell.

## Data availability
The authors declare that all data supporting the findings of this study are available within the paper or supplementary information files. Source data are provided with this paper. Source data are provided with this paper.

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

## Acknowledgements

This research was supported by National Key R&D Program of China (2020YFA0406202) (X.R.X.), National Natural Science Foundation of China (22401016 (C.Y.Y.), 22090042 (X.R.X.), 52122408 (H.H.W.)), Post-doctoral Fellowship Program of CPSF under Grant Number GZB20240058 (C.Y.Y.), China Postdoctoral Science Foundation (2023M740210) (C.Y.Y.), and the Fundamental Research Funds for the Central Universities, China (FRF-EYIT-23-03 (K.L.), FRF-IDRY-23-020 (Y.L.C.)). The synchrotron radiation experiments were performed at the BL44B2 and BL02B2 (Proposal numbers: 2023B1810, 2024A1066) of SPring-8 with the approval of the Japan Synchrotron Radiation Research Institute (JASRI). Neutron diffraction work was carried out at the WAND², Spallation Neutron Source (SNS) (Proposal No. 2023B32019.1), which is the U.S. Department of Energy (DOE) user facility at the Oak Ridge National Laboratory, sponsored by the Scientific User Facilities Division, Office of Basic Energy Sciences and a general-purpose powder dif-fractometer (GPPD) (Proposal No. P1823072700003) (90° bank) at the China Spallation Neutron Source (CSNS), Dongguan, China.

## Author contributions

Conceptualization: C.Y.Y., X.R.X. Methodology: H.H.W., X.C., M.S.Z., Q.Z. Investigation: C.Y.Y., H.H.Z., Q.H.Z., Y.J.C., L.G., M.D.F., Y.C., K.A., L.H.H., K.K., S.K., Z.Y.Q., Y.L.C., Q.L., J.X.D. Visualization: C.Y.Y., H.H.W. Funding acquisition: C.Y.Y., H.H.W., X.R.X. Project administration: C.Y.Y., X.R.X. Supervision: X.R.X. Writing—original draft: C.Y.Y. Writing—review & editing: H.H.W., K.L., X.R.X.

## Competing interests

The authors declare no competing interests.
