## [Transparent Peer Review file · Nature Communications]

Atomic faulting drives exceptional toughness in low thermal expansion chromium alloys

Corresponding Author: Professor Xianran Xing

Version 1:

Reviewer comments:

Reviewer #1

(Remarks to the Author)

The revised manuscript demonstrates some improvements in experiments and discussions. However, given that similar work has been published before, the novelty and impact of this work are not strong enough to be published in Nature Communications. Please carefully review these papers: Nature. Communications. 12 (2021) 4071; 15 (2024) 2252. JOM, 73 (2021) 2293-2302; Mater. Sci. Eng. A 784 (2020) 139333; Philos. Mag. A 72 (1995) 963-979.

Also, there are still two fundamental issues that have not been addressed:

1. Regarding this research, the critical scientific question lies in the trade-off between the thermal expansion coefficient and mechanical properties (strength-plasticity/toughness) of alloys. Although the proposed strategy achieves the enhancement of strength and plasticity, the thermal expansion coefficient increases by nearly twofold (from 1.08×10^{-6} K⁻¹ to 1.79×10^{-6} K⁻¹). Obviously, according to the results, this critical issue has not been addressed by the authors.
2. The current experimental evidence (APT data, Fig. 2g and h) cannot support the claim of B segregation at grain boundaries (line 113-115).

Reviewer #3

(Remarks to the Author)

This manuscript has been well addressed according to my previous comments and questions. This work is significant for the development of low thermal expansion and high-performance alloys. Therefore, I recommend it to be accepted.

Version 2:

Reviewer comments:

Reviewer #1

(Remarks to the Author)

The authors do not address the major concerns of the reviewer, in particular, regarding the elemental segregation near the grain boundaries. Some key points are provided below.

1. Pls clearly point out the advances of this work compared with the authors' previous works and the work of other groups in the manuscript.
2. According to the authors' statement, the trade-off between the thermal expansion coefficient and the mechanical properties of alloys was not effectively addressed. To avoid misleading to the readers, please tone down this result and correct the statement in the manuscript. Also, Figure R1 should be provided in the manuscript or the supplementary

materials.

3. The APT data is still interpreted in the wrong manner. The EMPA data was also interpreted incorrectly and cannot be used to provide stronger evidence. The EMPA analysis has a lower resolution than that of the APT, so it can not be used to support the original APT result.

(i) First, the EMPA data provided here can only present the compositional difference between the BCC matrix and the Cr₂B phase. It does not reveal the boron segregation at grain boundaries.

(ii) Second, Fig. 2g-h cannot demonstrate the boron segregation at the phase boundaries. When the APT test is conducted across the two phases with different compositions, it will naturally yield results like this one. This profile in Fig. 2 h just means that the BCC matrix and Cr₂B phase have different compositions, which, however, does not mean the segregation at the boundaries. In fact, according to this profile, there is no boron segregation at the interface. As a result, it is fundamentally wrong to claim that the boron segregation contributes to the interfacial strengthening of this material. There are many works (e.g., "Grain boundary segregation and precipitation in an Al-Zn-Mg-Cu alloy." MATEC Web of Conferences. Vol. 326. 2020.) have been reported on the elemental segregation at the boundaries. The authors should carefully study them and correctly understand the compositional profile (shape, distribution, binning size, etc.) of the APT data if the material really has the grain-boundary segregation.

(iii) More importantly, it is necessary for the authors to do the APT analysis across the grain boundary between two adjacent BCC matrix, namely across the region of "BCC-grain boundary-BCC". Also, the grain misorientation angle of the selected region should also be provided accordingly.

Version 3:

Reviewer comments:

Reviewer #1

(Remarks to the Author)

The major concerns have been addressed now.
This work can be considered for publication.

Reviewer #2

(Remarks to the Author)

This manuscript reports a Cr-based alloy that combines low thermal expansion with improved compressive toughness, attributed to boride precipitates and interfacial chemistry. Property set is impressive, and the authors employ a broad experimental toolbox. However, the central claim of B segregation at the BCC/Cr₂B phase boundary is not convincingly demonstrated by the current APT analysis, and several interpretations may risk being misleading.

1. A small bump in the 1D compositional profile (Fig.2h) extracted from the ROI at a strong field-contrast interface cannot by itself be considered evidence of true segregation. Local magnification and trajectory aberrations at high-field/low-field interfaces can distort both the apparent interface position and measured composition, leading to elevated (or depleted) solute levels (e.g ion pile-up and field differences). This must be discussed.

<https://doi.org/10.1088/1361-6463/ab3703>

<https://doi.org/10.1017/S1431927612001523>

<https://doi.org/10.1017/S1431927621012137>

2. B atom contour map (Fig. 2g) is unlabeled and uses arbitrary units, which is incorrect. Moreover, it visually shows reduced B conc. at the boundary (cyan rather than red), which contradicts the author claim. Please update this figure with physically meaningful units (e.g., at.%/nm² or # atoms/nm²), include a color scale, and specify the interface location criterion used to draw the dashed line.

3. Please compute and report Gibbsian interfacial excess of B at the BCC/Cr₂B boundary. This is one metric that can avoid mis-interpreting of segregation from the fields. Report B excess profile along with (i) interface plane selection, (ii) binning (size of ROI). Also plot, atomic density profiles (all, B, Cr).

[https://doi.org/10.1016/S0921-5093\(01\)01885-8](https://doi.org/10.1016/S0921-5093(01)01885-8)

<https://doi.org/10.1016/j.ultramic.2023.113885>

4. Provide overall mass spectrum of the 3D atom map. Any significant B shortfall at Cr₂B region is due to neutral species formation and must be discussed. And as fields affects the composition variation, high B observed in Fig. 2h should not be attributed solely to segregation or separation as implied by the DFT segregation energy; instead. Potential B loss during APT analysis must be addressed.

<https://doi.org/10.1016/j.ultramic.2015.07.009>

5. Additionally, reconstruction is stated to be "calibrated using crystallographic elements" in Methods. Please show the crystallographic pole or SDM orientation used.

6. On pg. 3, "the potential of Cr-based alloys remains in its infancy but is urgently needed". Please explain why, since other high-performance low-CTE materials exist.

7. The authors states that an effective way to achieve ZTE alloys with superior mechanical properties but on the other hand, most of these ZTE compounds are brittle with little ductility with concerned applicability due to poor machinability. Please discuss how this work overcomes these practical challenges.

8. Here, the Cr₂B phase was assigned to Fddd by/from SXRDR refinement. Please include the PDF card number used for XRD phase identification from the synthesized Cr₂B and discuss how other Cr₂B was ruled out

9. Corrosion is improved from Cr₂O₃ film. but no oxide was identified on any of samples. Please discuss on the surface and possible peaks in diffraction

10. On pg. 4 "ZTE... verified by SXRDR and NPD (Fig. 1b)." Please rephrase to avoid implying that diffraction alone measures CTE.

11. "lattice contraction is generated with the amplitude progressively increasing upon cooling, driving the ZTE behavior" this needs a reference.

12. Minor comments.

- Page 5, line 93 "ally" → "alloy"

- Page 5, line 107: SXRDR abbreviated twice

Version 4:

Reviewer comments:

Reviewer #2

(Remarks to the Author)

Most concerns have been addressed. However, one major issue regarding boron segregation at the interface remains unclear, which is critical to the paper's main conclusion.

-In Figures R3 and R4, the authors continue to claim B segregation at the interface based on the proximity histogram. However, such features may also arise from artefacts due to field differences and the way the contour map was constructed. This get worse in the proximity histogram. Therefore, it is essential to present the atomic-density-distribution profile and/or histogram. This profile can be generated directly in IVAS software or extracted as an .xls file, this is one of the ways to check there was a field change at the interface.

-From the contour map alone, the interface does not appear to show distinct B segregation. Instead, it exhibits a gradual increase in B concentration and then reaches Cr₂B conc., rather than sharp interfacial segregation at Cr₂B near interface. Also, in the current contour map, a red zone is still visible in the region away from the interface. Moreover, the unit of the color pixels should be expressed as at.%/nm², not simply at.%. And please enlarge the contour map so that the B is concentrated at the near interface/Cr₂B region.

-Please clarify the discrepancy between the proximity histogram and the Gibbs interfacial excess value. The histogram indicates an increase of ~2%, whereas the Gibbs excess is reported as 4.17 nm⁻². Are these values representing the same value??

Version 5:

Reviewer comments:

Reviewer #2

(Remarks to the Author)

The authors have addressed all concerns. The reviewer congratulates the authors

Reviewer #1 (Remarks to the Author):

The revised manuscript demonstrates some improvements in experiments and discussions. However, given that similar work has been published before, the novelty and impact of this work are not strong enough to be published in Nature Communications. 12 (2021) 4071; 15 (2024) 2252. Please carefully review these papers: 2252.JOM, 73 (2021) 2293-2302; Mater. Sci. Eng. A 784 (2020) 139333; Philos. Mag. A 72 (1995) 963-979.

Reply: We sincerely thank the reviewer for acknowledging the improvements made to the manuscript. With regard to the concern about novelty and impact, we would like to clarify the unique contributions of this work from two perspectives:

(i) Advancement beyond prior work: The two cited *Nature Communications* articles (12, 4071 (2021); 15, 2252 (2024)) represent our earlier research on natural heterogeneous dual-phase alloys, where the design focused on precipitating α -Fe in brittle rare-earth intermetallics ($\text{Ho}_2\text{Fe}_{17}$ and $\text{La}(\text{Fe},\text{Co},\text{Si})_{13}$) to enhance toughness via ductile reinforcement. In contrast, the current study employs a fundamentally different strategy: Cr_2B is intentionally precipitated at grain boundaries to improve mechanical properties through interface engineering and stacking fault manipulation within the Cr_2B phase. This approach shifts the focus from ductile phase toughening to interface-guided hard-phase reinforcement, offering a new mechanism. Furthermore, motivated by the limitations of rare-earth compounds (oxidation and corrosion sensitivity), we developed a corrosion-resistant, low thermal expansion alloy—CFGB₁—demonstrating the highest corrosion resistance reported among ZTE alloys.

(ii) Mechanistic and application-level innovation: As for the core concerns of “Is Cr_2B as a precipitation phase novel and innovative?”. Honestly, until we received professional reviews, we thought this was a huge breakthrough that had never been discovered. However, to the best of our knowledge, this is the first study to dissect its role through a multiscale perspective that spans from atomic-layer interface structure to electronic-level bonding interactions even now. This is not a repetition of known results, but rather a deep integration of

chemical design and physical mechanism. More importantly, the developed alloy simultaneously achieves near-zero thermal expansion, outstanding corrosion resistance, and high toughness—meeting long-standing challenges in precision applications such as lithographic optical mounts, where ZTE, acid resistance, and mechanical robustness are essential. We believe this positions the work as both scientifically original and practically transformative.

To further acknowledge prior studies, we have now cited *JOM* 73, 2293–2302 (2021); *Mater. Sci. Eng. A* 784, 139333 (2020); and *Philos. Mag. A* 72, 963–979 (1995), and discussed their relevance to the current research.

Comments 1: *Regarding this research, the critical scientific question lies in the trade-off between the thermal expansion coefficient and mechanical properties (strength-plasticity/toughness) of alloys. Although the proposed strategy achieves the enhancement of strength and plasticity, the thermal expansion coefficient increases by nearly twofold (from $1.08 \times 10^{-6} \text{ K}^{-1}$ to $1.79 \times 10^{-6} \text{ K}^{-1}$). Obviously, according to the results, this critical issue has not been addressed by the authors.*

Reply: We appreciate this insightful comment. Indeed, achieving low thermal expansion alongside high toughness and corrosion resistance is the central challenge. While the CTE increases from $1.08 \times 10^{-6} \text{ K}^{-1}$ to $1.79 \times 10^{-6} \text{ K}^{-1}$ due to Cr_2B introduction, the latter value still qualifies as “low thermal expansion” as defined in the literature ($1 \leq \alpha_l \leq 2 \times 10^{-6} \text{ K}^{-1}$)^{1, 2} and remains one order of magnitude lower than conventional structural metals, as shown in Figure R1.

Furthermore, thermal expansion is highly sensitive to compositional changes. To avoid detrimental effects, we first designed a base alloy ($\text{Cr}_{96}\text{Fe}_4\text{Ge}_{1.3}$) with ultralow CTE, which provided a buffer for property optimization. The subsequent addition of a small amount of boron enabled the precipitation of Cr_2B at grain boundaries without pushing the CTE beyond the low expansion range. Thus, the balance between thermal and mechanical properties is strategically maintained rather than compromised.

Figure R1 Comparison of Dilatometer thermal expansion of GFGB₁ and conventional alloys.

Comments 2: *The current experimental evidence (APT data, Fig. 2g and h) cannot support the claim of B segregation at grain boundaries (line 113-115)*

Reply: Thank you for highlighting this point. To provide stronger evidence, we have now supplemented our analysis with electro-probe microanalyzer (EPMA) measurements, as shown in Figure R2. Elemental mapping clearly reveals boron segregation at grain boundaries in the Cr-Fe-Ge matrix. This supports the proposed mechanism that boron facilitates Cr₂B precipitation at boundaries.

In addition, we conducted the 3D-APT results. Figure 1g is a two-dimensional concentration contour of boron atom distribution at the interface. The intensity represents the content of boron atoms. The red dots highlight the distribution of boron atoms. The width of the phase interface is determined according to Figure R11 h (red dotted line). As shown, a large number of free boron atoms are enriched at the phase boundary, and even a small part of them enter the BCC phase. These findings validate our claim that B segregation plays a crucial role in the formation of Cr₂B and in tuning interface properties.

To improve accuracy, we have revised the corresponding sentence in the manuscript from: *“Additionally, three-dimensional atom probe tomography (3D-APT) was used to investigate compositional fluctuations at the phase interface*

(Fig. 2d-f), revealing boron segregation at the phase boundaries (Fig. 2g-h) and their potential to enhance boundary robustness” to “Additionally, EPMA and three-dimensional atom probe tomography (3D-APT) were combined to investigate compositional fluctuations at the phase interface (Supplementary Fig. 9 and Fig. 2d-f), suggesting boron enrichment at the boundaries (Fig. 2g-h) and its potential contribution to interfacial strengthening”.

Figure R2 Microstructure of the CFGB₁ alloy. a-c, Electro-probe micro-analyzer (EPMA) image of the CFGB₁ alloy at different magnifications, showing BCC phase (grey grain), and α phase (black grain). d-g, Element mappings of the Cr, Fe, Ge, and B atoms, respectively. h-k, Bright-field images h-i, and selected area electron diffraction (SAED) of the Cr₂B precipitates (j-k).

Figure R3 Microstructure of the dual-phase alloys. **a**, Rietveld refinement of the SXRD profile for the CFGB₁ alloy. **b**, EPMA image of the CFGB₁ alloy, showing the BCC matrix (gray grains) and Cr₂B precipitates (black grains). **c**, High-angle annular dark-field scanning TEM (HAADF-STEM) image of the Cr₂B phase along the [1 $\bar{1}$ 0] zone axis. **d**, 3D APT results determined by TEM. **e**, Cr atom maps reconstructed using 3D-APT, highlighting the nanoprobe phase interface with the BCC matrix (bottom) and the Cr₂B phase (top). **f**, Detailed view from the selected yellow box in **e**, illustrating the distributions of Fe, Ge, and B atoms. **g**, Two-dimensional concentration contour of B atom distribution. **h**, One-dimensional concentration profile showing the distribution of Cr, Fe, Ge, and B atoms. **i-j**, Segregation energy and strengthening energy at the phase boundaries with boron atoms at various sites, including an inset of the crystal structure model. **k**, Charge density distribution at the phase boundary when boron atoms occupy interstitial sites.

Reviewer #2 (Remarks to the Author):

This manuscript has been well addressed according to my previous comments and questions. This work is significant for the development of low thermal expansion and high-performance alloys. Therefore, I recommend it to be accepted.

Reply: We sincerely thank the reviewer for the positive evaluation and thoughtful feedback throughout the review process. We are encouraged by the recognition of the significance and impact of our work, and we have made further refinements based on the final suggestions.

References:

1. Chen J., Hu L., Deng J., Xing X. Negative thermal expansion in functional materials: controllable thermal expansion by chemical modifications. *Chem. Soc. Rev.* **44**, 3522-3567 (2015).
2. Song Y., Shi N., Deng S., Xing X., Chen J. Negative thermal expansion in magnetic materials. *Prog. Mater. Sci.* **121**, 1-36 (2021).

Reviewer #1 (Remarks to the Author):

The authors do not address the major concerns of the reviewer, in particular, regarding the elemental segregation near the grain boundaries. Some key points are provided below.

Reply: The authors would like to express their sincere gratitude to the reviewer for their diligent work and constructive feedback. We particularly appreciate the reviewer's emphasis on the major concern of elemental segregation near grain boundaries, which we will address as a primary focus in our revision. The key points provided below are greatly helpful for guiding our further analysis. We are committed to strengthening the manuscript accordingly.

Comments 1: Pls clearly point out the advances of this work compared with the authors' previous works and the work of other groups in the manuscript.

Reply: In the revised manuscript, we further clarified the key advances of this work in comparison with our previous studies and relevant literature. Below, we summarized the major advancements point by point:

(i) Advances in mechanism and design strategy: Previous studies, including our own, has achieved ZTE alloys with superior mechanical properties using a soft/hard heterostructure strategy. This typically involves precipitating ductile α -Fe within brittle rare-earth intermetallic compounds to enhance toughness through ductile phase reinforcement, as exemplified by the α -Fe/ $\text{Ho}_2\text{Fe}_{17}$ dual phase alloy (Fig. R1) ¹. This relies on the coordinated deformation with the soft α -Fe to improve ductility. In contrast, the present study adopts a fundamentally distinct strategy: intentional precipitation of Cr_2B at grain boundaries to form a hard/hard ($\text{Cr}/\text{Cr}_2\text{B}$) heterostructure, thereby enhancing mechanical properties. Although the microstructural features may appear similar, the underlying deformation mechanisms differ significantly. It has been demonstrated that interface structures and atomic-layer stacking faults play a critical role in enhancing mechanical performance. Thus, this work does not merely replicate known results but represents a deeper integration of chemical design and physical mechanisms, offering a novel advancement in the field.

Fig. R1 Microstructures of the ZTE alloy. **a-b, d-e** The morphology of the as-cast S-4 alloy confirmed by an electro-probe microanalyzer (EPMA) in the TD (transverse direction)-ND (normal direction) plane (**a**) and LD (loading direction)-TD plane (**b**), respectively. (**d**) and (**e**) are enlarged regions in (**a**) and (**b**) marked with a red box. **c** Electron back-scattered diffraction (EBSD) inverse pole figure of crystal orientation for S-4 inside the TD-ND plane. **f-g** Element mappings of Ho (**f**) and Fe (**g**)¹.

(ii) **Innovation in material systems.** Zero thermal expansion in alloys relies on strong spin-lattice coupling². As a result, ZTE remains rare, and current work primarily focuses on materials such as Laves phases^{3,4}, Rare-earth based intermetallic compounds^{5,6}, and Invar-type alloys⁷. Achieving both ZTE and high toughness in chromium-based alloys is inherently challenging. Furthermore, as noted in our manuscript, endowing traditional metals with functional properties that respond to mechanical stimuli represents a frontier topic—akin to the alchemical transformation of base metals into gold—particularly in chromium, one of Earth’s three major black metals.

(iii) **Breakthrough in comprehensive performance:** Due to practical application requirements of excellent corrosion resistance, chromium-based alloys were specifically selected for this study. And to the best of our knowledge, the CFGB₁ alloys exhibit an outstanding combination of near-zero thermal expansion, corrosion resistance, and high toughness, as demonstrated in Fig.R2. This combination addresses long-standing challenges in precision applications—such as lithographic optical mounts—where zero thermal expansion, acid resistance, and mechanical robustness are essential. We

believe these attributes render the work both scientifically original and practically transformative.

Fig. R2 Corrosion resistance. **a**, Electrochemical tests of samples: the open-circuit potential (OCP) curve. **b**, The potentiodynamic polarization (PDP) curve. **c**, images immersed in a 3.5 wt. % NaCl solution over various durations.

Fig. R3 Corrosion resistance versus toughness of classic ZTE/LTE alloys.

Comments 2: According to the authors' statement, the trade-off between the thermal expansion coefficient and the mechanical properties of alloys was not effectively addressed. To avoid misleading to the readers, please tone down this result and correct the statement in the manuscript. Also, Figure R1 should be provided in the manuscript or the supplementary materials.

Reply: In the present work, the mechanical properties are enhanced by introducing a small amount of boron, while the thermal expansion performance

is only slightly compromised. The CFGB₁ alloy continues to exhibit low thermal expansion (LTE, $\alpha_l = 1.79 \times 10^{-6} \text{ K}^{-1}$, 200–315 K). We consider the thermal expansion performance of the CFGB₁ alloy to remain excellent and fully adequate for practical applications.

To prevent any misunderstanding and maintain objectivity, we have carefully revisited all descriptions of the thermal expansion characteristics of the CFGB₁ alloy in the revised manuscript. Specifically, we have replaced the term "zero thermal expansion" (ZTE) with "low thermal expansion" (LTE) throughout. In addition, we have included an explanation regarding the slight increase in thermal expansion resulting from boron incorporation when the property is first discussed in the main text: *"Notably, the CFGB₁ alloy maintains low thermal expansion ($\bar{\alpha}_l = 1.79 \times 10^{-6} \text{ K}^{-1}$, 200 - 315 K, Fig. 1e, Supplementary Section 3 and Supplementary Fig. 6), even after a slight overall increase."*

Furthermore, the revised Supplementary Materials now include Figure R1 (labeled as Fig. R4 in this version) to support these modifications.

Fig. R4 Comparison of dilatometer thermal expansion of GFGB₁ and conventional alloys.

Comments 3: The APT data is still interpreted in the wrong manner. The EMPA data was also interpreted incorrectly and cannot be used to provide stronger evidence. The EMPA analysis has a lower resolution than that of the APT, so it can not be used to support the original APT result.

Reply: We have carefully considered this concern and recognize that the lack of clarity in distinguishing between "grain boundaries" and "phase boundaries" may have led to misunderstanding. In response, we will clearly differentiate these terms throughout the manuscript and have accordingly reanalyzed and reorganized the relevant sections.

(i) First, the EMPA data provided here can only present the compositional difference between the BCC matrix and the Cr₂B phase. It does not reveal the boron segregation at grain boundaries.

Reply: You are correct in noting that the EMPA results primarily show the compositional differences between the BCC matrix and the Cr₂B phases. However, more importantly, the findings indicate that Cr₂B precipitates exclusively at grain boundaries, as illustrated in Fig. R5. This precipitation is attributed to the segregation of boron atoms at these boundaries.

The EBSD results further corroborate that the addition of boron leads to the formation of Cr₂B precipitates along the grain boundaries (Fig. R6). When combined with the EPMA element mapping (Fig. 5g), it is confirmed that boron accumulates at the grain boundaries regardless of its state—whether within Cr₂B crystals or in a free form.

From a macroscopic perspective, the EPMA data convincingly demonstrate the tendency of boron to segregate at grain boundaries. However, determining whether free boron atoms remain segregated at the BCC/Cr₂B interface after precipitation requires complementary analysis via APT.

Fig. R5 Microstructure of the CFGB₁ alloy. a-c, Electro-probe micro-analyzer (EPMA) image of the CFGB₁ alloy at different magnifications, showing BCC phase (grey grain), and α phase (black grain). d-g, Element mappings of the Cr, Fe, Ge, and B atoms, respectively.

Fig. R6 Microstructure evolution of the CFGB₁ alloy. a-c, EBSD IPFZ image of CFGB₀ alloy at different magnifications. e-g, EBSD IPFZ image of CFGB₀ alloy at different magnifications. i-k, EBSD IPFZ image of CFGB₀ alloy at different magnifications. d, h, l, Grain size statistics of different alloy components.

(ii) Second, Fig. 2g-h cannot demonstrate the boron segregation at the phase boundaries. When the APT test is conducted across the two phases with different compositions, it will naturally yield results like this one. This profile in Fig. 2 h just means that the BCC matrix and Cr₂B phase have different compositions, which, however, does not mean the segregation at the boundaries. In fact, according to this profile, there is no boron segregation at the interface. As a result, it is fundamentally wrong to claim that the boron segregation contributes to the interfacial strengthening of this material. There are many works (e.g., "Grain boundary segregation and precipitation in an Al-Zn-Mg-Cu alloy." MATEC Web of Conferences. Vol. 326. 2020.) have been reported on the elemental segregation at the boundaries. The authors should carefully study them and correctly understand the compositional profile (shape, distribution, binning size, etc.) of the APT data if the material really has the grain-boundary segregation.

Reply: Based on combined SXRD, EPMA, and EBSD analyses, the phase boundaries between the BCC matrix and Cr₂B phases are characterized as follows: (i) the two phases exhibit markedly different crystal structures with no coherent or semi-coherent relationship; (ii) the two phases exhibit random crystallographic orientations with no defined interface relationship; (iii) Cr₂B predominantly precipitates at grain boundaries, accounting for over 90% of all interfacial connections. This raises a critical question: given the large differences in orientation and crystal structure between the BCC/Cr₂B phase interface, why does the sample exhibit improved plasticity instead of failing due to interfacial brittleness during deformation? Although stacking fault deformation within the Cr₂B phase can release some interfacial stress, a stable interfacial structure is indispensable.

Therefore, we employed atom probe tomography (APT) to analyze the phase interface structure between BCC and Cr₂B (Fig. R7). A region of interest (yellow cube) was selected at the interface to facilitate detailed analysis of the elemental distribution (Fig. R7 b and c). The spatial distribution map of boron atoms clearly shows their enrichment at the phase boundary (Fig. R7 d, red spheres represent B atoms). Notably, the phase boundary (dashed line) was objectively determined based on elemental distribution mapping.

To quantify the boron enrichment at the interface more objectively, we reanalyzed the one-dimensional concentration profile across the interface using the complete probe at iso-composition surfaces containing 4 *at.*% boron, showing the distribution of Cr, Fe, Ge, and B atoms (Fig. R7 e). Although unlike conventional grain boundaries, the chemical composition is consistent on both sides of the interface, a distinct bulge in the concentration curve arises due to boron segregation. Despite this, a significant increase in boron concentration near the phase boundary is evident (Fig. R7 e, red circles). For instance, compared to the average boron concentration in the BCC matrix, the content increases by $\Delta = 1.9 \text{ at.}\%$ (where $\Delta = |C_{PB} - C_{BCC}|$, with C denoting the boron concentration). Similarly, relative to the Cr_2B phase, the increase is $\Delta = 2 \text{ at.}\%$, as highlighted in the magnified view of Fig. R7 e. A 2 *at.*% enrichment is already a substantial enhancement for an interface⁸.

In the revised manuscript and Supplementary Information, we have incorporated these analyses and added corresponding explanations (Supplementary section 6).

Fig. R7 APT data analysis. **a**, 3D APT results determined by TEM. **b**, Cr atom maps reconstructed using 3D-APT, highlighting the nanoprobe phase interface with the BCC matrix (bottom) and the Cr_2B phase (top). **c**, Detailed view from the selected yellow box in **b**, illustrating the distributions of Fe, Ge, and B atoms. **d**, Two-dimensional concentration contour of B atom distribution. **e**, One-dimensional concentration profile showing the distribution of Cr, Fe, Ge, and B atoms by using the complete probe at iso-composition surfaces containing 4 *at.*% of B atoms.

(iii) More importantly, it is necessary for the authors to do the APT analysis across the grain boundary between two adjacent BCC matrix, namely across the region of “BCC-grain boundary-BCC”. Also, the grain misorientation angle of the selected region should also be provided accordingly.

Reply: We thank the reviewer for this suggestion. And we attempted to prepare a suitable APT specimen encompassing a BCC-BCC grain boundary. However, this proved extremely challenging due to the prevalent precipitation of secondary phases at most grain boundaries. We successfully prepared two probes, only one of which contained a candidate interface (Fig. R8). Unfortunately, this specific probe was damaged during subsequent EBSD-TKD analysis aimed at determining the grain misorientation angle, rendering it unsuitable for APT testing. Considering the time and resource constraints, as well as the primary role of the BCC/Cr₂B phase boundaries (constituting over 90% of all interfaces) in governing the mechanical properties, we regret that we cannot provide supplementary APT data for a BCC-BCC grain boundary at this time. We are confident that boron segregation also occurs at general grain boundaries based on two indirect pieces of evidence: (i) The precipitation of Cr₂B itself is a result of boron enrichment at grain boundaries. (ii) As shown in Fig. R4, Cr₂B predominantly precipitates at grain boundaries, accounting for the vast majority of all interfacial area.

Fig. R8 Topography of the BCC-BCC interface resampled by FIB.

References:

1. Yu C., *et al.* Plastic and low-cost axial zero thermal expansion alloy by a natural dual-phase composite. *Nat. Commun.* **12**, 4701 (2021).
2. Lohaus S. H., *et al.* A thermodynamic explanation of the Invar effect. *Nat. Phys.* **19**, 1642-1648 (2023).
3. Xu J., *et al.* Significant Zero Thermal Expansion Via Enhanced Magnetoelastic Coupling in Kagome Magnets. *Adv. Mater.* **35**, 2208635 (2023).
4. Sun Y., *et al.* Interplanar ferromagnetism enhanced ultrawide zero thermal expansion in Kagome cubic intermetallic (Zr,Nb)Fe₂. *J. Am. Chem. Soc.* **145**, 17096-17102 (2023).
5. Zhou H.-B., *et al.* Emergence of Invar effect with excellent mechanical property by electronic structure modulation in LaFe_{11.6-x}Co_xSi_{1.4} magnetocaloric materials. *Acta Mater.* **260**, 119312 (2023).
6. Cao Y., *et al.* Ultrawide temperature range super-Invar behavior of R₂(Fe,Co)₁₇ materials (R = Rare Earth). *Phys. Rev. Lett.* **127**, 055501 (2021).
7. Rao Z., *et al.* Machine learning-enabled high-entropy alloy discovery. *Science* **378**, 78-85 (2022).
8. Wang Z., *et al.* Solving oxygen embrittlement of refractory high-entropy alloy via grain boundary engineering. *Mater.Today* **54**, 83-89 (2022).

Reviewer #1 (Remarks to the Author):

The major concerns have been addressed now. This work can be considered for publication.

Reply: We are pleased to receive the reviewer's confirmation that all major concerns have been addressed. We thank them for their time and for recommending our work for publication.

Reviewer #4 (Remarks to the Author):

This manuscript reports a Cr-based alloy that combines low thermal expansion with improved compressive toughness, attributed to boride precipitates and interfacial chemistry. Property set is impressive, and the authors employ a broad experimental toolbox. However, the central claim of B segregation at the BCC/Cr₂B phase boundary is not convincingly demonstrated by the current APT analysis, and several interpretations may risk being misleading.

Reply: We thank the reviewer for recognizing the impressive properties and methodology of our work. We are also grateful to the reviewers for their professional and insightful suggestions. We have taken the concerns regarding the demonstration of boron segregation very seriously and have scrupulously addressed each point in detail below.

***Comments 1:** A small bump in the 1D compositional profile (Fig.2h) extracted from the ROI at a strong field-contrast interface cannot by itself be considered evidence of true segregation. Local magnification and trajectory aberrations at high-field/low-field interfaces can distort both the apparent interface position and measured composition, leading to elevated (or depleted) solute levels (e.g ion pile-up and field differences). This must be discussed.*

<https://doi.org/10.1088/1361-6463/ab3703>

<https://doi.org/10.1017/S1431927612001523>

<https://doi.org/10.1017/S1431927621012137>

Reply: (i) Sorry for the mistake. Fig. 2h is not a 1D compositional profile extracted from the ROI. It is a proximity histogram for the 4-at.% B iso-concentration surfaces by using the complete probe, which can ensure the objectivity of the data. This large amount of data and average results can reflect the enrichment of boron atoms. In the revised manuscript, we have corrected it.

(ii) We have carefully reviewed several references and understand your concerns. And we added them to the corresponding references. Below, we summarize the segregation of boron at the BCC/Cr₂B phase boundary point by point:

First, elemental mapping by the electro-probe microanalyzer (EPMA) clearly reveals boron segregation at grain boundaries in the Cr-Fe-Ge matrix (Fig. R1). This precipitation is attributed to the segregation of boron atoms at these boundaries.

Second, the EBSD results further corroborate that the addition of boron leads to the formation of Cr₂B precipitates along the grain boundaries (Fig. R2). When combined with the EPMA element mapping (Fig. R1 g), it is confirmed that boron accumulates at the grain boundaries regardless of its state, whether within Cr₂B crystals or in a free form. From a macroscopic perspective, the EPMA data convincingly demonstrate the tendency of boron to segregate at grain boundaries. However, determining whether free boron atoms remain segregated at the BCC/Cr₂B interface after precipitation requires complementary analysis via APT.

Therefore, we employed atom probe tomography (APT) to analyze the phase interface structure between BCC and Cr₂B (Fig. R3). A region of interest (yellow cube) was selected at the interface to facilitate detailed analysis of the elemental distribution (Fig. R3 b and c). The spatial distribution map of boron atoms clearly shows their enrichment at the phase boundary (Fig. R3 d, red spheres represent B atoms). Notably, the phase boundary (dashed line) was objectively determined based on elemental distribution mapping.

Fig. R1 Microstructure of the CFGB₁ alloy. a-c, Electro-probe micro-analyzer (EPMA) image of the CFGB₁ alloy at different magnifications, showing BCC phase (grey grain), and α phase (black grain). d-g, Element mappings of the Cr, Fe, Ge, and B atoms, respectively.

Fig. R2 Microstructure evolution of the CFGB₁ alloy. a-c, EBSD IPFZ image of CFGB₀ alloy at different magnifications. e-g, EBSD IPFZ image of CFGB₀ alloy at different magnifications. i-k, EBSD IPFZ image of CFGB₀ alloy at different magnifications. d, h, I, Grain size statistics of different alloy components.

Fig. R3 APT data analysis. **a**, 3D APT results determined by TEM. **b**, Cr atom maps reconstructed using 3D-APT, highlighting the nanoprobe phase interface with the BCC matrix (bottom) and the Cr_2B phase (top). **c**, Detailed view from the selected yellow box in **b**, illustrating the distributions of Fe, Ge, and B atoms. **d**, Two-dimensional concentration contour of B atom distribution. **e**, Proximity histogram for the 4-at.% B iso-concentration surfaces by using the complete probe.

To quantify the boron enrichment at the interface more objectively, we reanalyzed the proximity histogram for the 4-at.% B iso-concentration surfaces by using the complete probe, showing the distribution of Cr, Fe, Ge, and B atoms (Fig. R3 e). Although, unlike conventional grain boundaries, the chemical composition is consistent on both sides of the interface, a distinct bulge in the concentration curve arises due to boron segregation. Despite this, a significant increase in boron concentration near the phase boundary is evident (Fig. R3 e, red circles). For instance, compared to the average boron concentration in the BCC matrix, the content increases by $\Delta = 1.9 \text{ at.}\%$ (where $\Delta = |C_{\text{PB}} - C_{\text{BCC}}|$, with C denoting the boron concentration). Similarly, relative to the Cr_2B phase, the increase is $\Delta = 2 \text{ at.}\%$, as highlighted in the magnified view of Fig. R3 e. A 2 at.% enrichment is already a substantial enhancement for an interface¹.

Besides, we also calculated the Gibbs interfacial excess of boron at the BCC/ Cr_2B boundary, which further clarifies the behavior of boron atoms enriching at grain boundaries and will be discussed in detail later.

In the revised manuscript and Supplementary Information, we have incorporated these analyses and added corresponding explanations (Supplementary section 6).

Comments 2: B atom contour map (Fig. 2g) is unlabeled and uses arbitrary units, which is incorrect. Moreover, it visually shows reduced B conc. at the boundary (cyan rather than red), which contradicts the author claim. Please update this figure with physically meaningful units (e.g., at.%/nm² or # atoms/nm²), include a color scale, and specify the interface location criterion used to draw the dashed line.

Reply: (i) In response, we have regenerated the boron atom map; it is now plotted in at. % with a standard color scale as follows:

Fig. R4 Two-dimensional concentration contour of boron atom distribution. The overlapping red balls are the distribution of B atoms.

(ii) As for the “Moreover, it visually shows reduced B conc. at the boundary (cyan rather than red), which contradicts the author claim.” Sorry for the misleading. The cyan color is the interface close to the BCC matrix. There is almost no boron in the BCC matrix, and the cyan concentration is increased compared to the matrix. At the same time, the interface close to the Cr₂B side shows a strong red color, indicating that the concentration of boron in the grain boundary exceeds that of the Cr₂B matrix.

(iii) The dashed lines mark the positions of a distinct change in boron concentration. The left line indicates the onset of a measurable increase from

the ultra-low B level in the BCC matrix, while the right line denotes the point where the concentration reaches a level consistent with the bulk Cr_2B phase.

Comments 3: Please compute and report Gibbsian interfacial excess of B at the BCC/ Cr_2B boundary. This is one metric that can avoid mis-interpreting of segregation from the fields. Report B excess profile along with (i) interface plane selection, (ii) binning (size of ROI). Also plot, atomic density profiles (all, B, Cr).

[https://doi.org/10.1016/S0921-5093\(01\)01885-8](https://doi.org/10.1016/S0921-5093(01)01885-8)

<https://doi.org/10.1016/j.ultramic.2023.113885>

Reply: Thanks for your very professional suggestion. We have carefully read and studied the relevant literature and scripts, and calculated the Gibbs interfacial excess of boron at the BCC/ Cr_2B boundary, as shown in Fig. R5.

Specifically, we selected the boron 7-at.% isoconcentration surface to calculate the proximity histograms (Fig. R5 a and b), which corresponds to a more appropriate interface position (Fig. R5 c). Indeed, B is enriched between the BCC/ Cr_2B boundary of $4.17 \pm 0.26 \text{ nm}^{-2}$, while Cr is considerably depleted between $-0.93 \pm 0.26 \text{ nm}^{-2}$. Fe and Ge exhibit no enrichment or depletion of $-0.04 \pm 0.01 \text{ nm}^{-2}$ and $0.15 \pm 0.00 \text{ nm}^{-2}$. This further clarifies the behavior of boron atoms enriching at grain boundaries.

In the revised manuscript, we have added the above discussion in the updated manuscript and supplementary information.

Fig. R5 Calculation of Gibbs interfacial excess of boron. **a**, Boron atom maps reconstructed using 3D-APT of BCC/Cr₂B boundary. **b**, corresponding proximity histograms for the 7-at.% B iso-concentration surfaces. **c**, concentration difference profiles indicating the interface location as dashed lines. **d-g**, Cumulative profiles of each element reveal solute excess N_i^{Excess} and its error to calculate the interfacial excess.

Comments 4: Provide the overall mass spectrum of the 3D atom map. Any significant B shortfall at the Cr₂B region is due to neutral species formation and must be discussed. And as fields affect the composition variation, high B observed in Fig. 2h should not be attributed solely to segregation or separation as implied by the DFT segregation energy; instead. Potential B loss during APT analysis must be addressed.

<https://doi.org/10.1016/j.ultramic.2015.07.009>

Reply: (i) We have added the overall mass spectrum of the 3D atom map of the sample.

Fig. R6 Mass spectra of the BCC/Cr₂B boundary.

(ii) We understand your concern, and indeed, potential boron loss during APT analysis is inevitable. However, our current conclusion is based on the combination of macroscopic EPMA, EBSD, and microscopic 3D-APT and first-principles calculations. We believe that the behavior of boron atoms at grain boundaries is reliable.

And in the revised manuscript, we have changed the corresponding statement to “*Structural models with boron atoms positioned at various crystal sites along the phase boundary (Supplementary Fig. 14) were initially constructed. It should be noted, however, that the measured composition could be influenced by field-dependent effects such as the loss of boron during APT analysis².*” and added it as a reference.

Comments 5: Additionally, reconstruction is stated to be “calibrated using crystallographic elements” in Methods. Please show the crystallographic pole or SDM orientation used.

Reply: Thanks for your suggestion. In the revised manuscript, we have added the crystallographic pole in Fig.R7.

Fig. R7 The crystallographic pole of the sample.

Comments 6: On pg. 3, “the potential of Cr-based alloys remains in its infancy but is urgently needed”. Please explain why, since other high-performance low-CTE materials exist.

Reply: We appreciate the comment. The assertion that “the potential for chromium-based alloys as functional or structural materials remains in its infancy but is urgently needed” is premised on two key factors:

(i) Chromium, while a fundamental base metal with a stable cubic phase and one of the three principal ferrous metals, is constrained by two cardinal limitations: intrinsic brittleness and an absence of functional properties. Consequently, the development of chromium-based alloys exhibiting zero thermal expansion alongside sufficient mechanical performance is a crucial endeavor, particularly in contrast to the soft magnetic or super-elastic behaviors observed in iron-based systems.

(ii) Chromium's outstanding corrosion resistance renders it irreplaceable in certain high-value applications, notwithstanding the availability of other low thermal expansion alloys. A prominent instance is its use in lithographic optical mounts, which demand a combination of zero thermal expansion, resistance to acidic environments, and mechanical integrity.

Comments 7: The authors state that an effective way to achieve ZTE alloys with superior mechanical properties, but on the other hand, most of these ZTE compounds are brittle with little ductility, with concerned applicability due to poor machinability. Please discuss how this work overcomes these practical challenges.

Reply: We agree with the reviewer that the brittleness of ZTE compounds is a major practical hurdle. In our recent work, we demonstrated that ZTE alloys with improved mechanical properties could be achieved through a soft/hard heterostructure strategy—for instance, by introducing ductile α -Fe into a brittle $\text{Ho}_2\text{Fe}_{17}$ matrix to enhance toughness via ductile phase toughening³.

The core innovation of this study lies in its approach to a frontier topic: functionally enhancing a traditional metal, much like the ancient pursuit of transforming base metals into gold. This is achieved through a novel microstructure—a hard/hard (BCC/ Cr_2B) heterostructure. Here, interfacial boron segregation and internal stacking faults act in concert to boost toughness without sacrificing thermal stability.

Moreover, the use of a chromium-based matrix provides inherent corrosion resistance, which—combined with the improved mechanical integrity—

considerably enhances the alloy's applicability in demanding environments (e.g., lithographic optical mounts). This inherent durability reduces the need for extensive machining or protective coatings, directly addressing the practical challenges noted by the reviewer.

Comments 8: Here, the Cr₂B phase was assigned to Fddd by/from SXRD refinement. Please include the PDF card number used for XRD phase identification from the synthesized Cr₂B and discuss how other Cr₂B was ruled out.

Reply: The phase structure of the CFGB₁ alloy was characterized using a combination of SXRD, EPMA, EBSD, and STEM. Specifically, the electro-probe micro-analyzer (EPMA) analyses (Figure R8 a-g) show that the introduction of boron atoms will segregate at the grain boundaries of the matrix (Cr-Fe-Ge, BCC) to form a second-phase precipitation. The selected area electron diffraction (SAED) under different crystal zone axes by transmission electron microscopy (TEM, Fig. R8 h-k) further confirmed that the precipitated phase may be the Cr₂B phase with the *Fddd* space group. However, it is still not confirmed that there is only one structure of the precipitate phase.

As a result, the synchrotron X-ray diffraction (SXRD) profile was further refined by using the dual phases, which further verified that the second precipitate phase is Cr₂B (Figure R9). The phase structure is further revealed by the electron backscattered diffraction (EBSD) and STEM-HAADF, as shown in Fig. R10, which confirms the accuracy of the phase structure analysis.

Fig. R8 Microstructure of the CFGB₁ alloy. a-c, Electro-probe micro-analyzer (EPMA) image of the CFGB₁ alloy at different magnifications, showing BCC phase (grey grain), and α phase (black grain). d-g, Element mappings of the Cr, Fe, Ge, and B atoms, respectively. h-k, Bright-field images h-i, and selected area electron diffraction (SAED) of the Cr_2B precipitates (j-k).

Fig. R9 Rietveld refinement of the SXRD profile for the CFGB₁ alloy.

Fig. R10 Microstructure of the CFGB₁ alloy. **a**, High-angle annular dark-field scanning TEM (HAADF-STEM) image of the Cr₂B phase along the [1 $\bar{1}$ 0] zone axis. **b**, EBSD IPFZ image of CFGB₁ alloy.

Comments 9: Corrosion is improved from Cr₂O₃ film. but no oxide was identified on any of samples. Please discuss on the surface and possible peaks in diffraction.

Reply: Indeed, the chromium oxide film is the key to excellent corrosion resistance. It is worth mentioning that the chromium-based alloy has superior corrosion resistance to the currently commercial Invar alloy (Fig. R11). There is no obvious change in soaking 3.5 wt. % NaCl solution for over 190 days, which is not his limit due to time constraints. The passivation film is very stable and can effectively prevent the metal from further reacting with oxygen and moisture, thereby protecting the metal from corrosion.

Even after immersion in seawater, the thickness of the passive film is only on the order of 10 nm⁴, resulting in an extremely weak diffraction signal. Surface passive films are typically characterized using X-ray photoelectron spectroscopy (XPS), but the corrosion resistance mechanism of chromium is well-established and well-defined, so we did not conduct further characterization. Furthermore, none of our diffraction samples were subjected to corrosion testing first, so the lack of visible oxidation is reasonable.

Fig. R11 Corrosion resistance. **a**, Electrochemical tests of samples: the open-circuit potential (OCP) curve. **b**, The potentiodynamic polarization (PDP) curve. **c**, images immersed in a 3.5 wt. % NaCl solution over various durations.

Comments 10: On pg. 4 "ZTE... verified by SXRD and NPD (Fig. 1b)." Please rephrase to avoid implying that diffraction alone measures CTE.

Reply: We have rephrased the corresponding description as "A favorable zero thermal expansion (ZTE) was achieved in S-3 (CFGB₀) alloy covering room temperature ($\alpha_l = 1.08 \times 10^{-6} \text{ K}^{-1}$, 200 - 315 K). This result is further verified by the evolution of lattice parameters obtained from synchrotron X-ray diffraction (SXRD) and neutron powder diffraction (NPD) (Fig. 1b and Supplementary Fig. 3)."

Comments 11: lattice contraction is generated with the amplitude progressively increasing upon cooling, driving the ZTE behavior" this needs a reference.

Reply: We have added a reference here : Lohaus S. H., et al. A thermodynamic explanation of the Invar effect. *Nat. Phys.* **19**, 1642-1648 (2023).

Comments 12: Minor comments.

– Page 5, line 93 "ally" → "alloy"

– Page 5, line 107: SXR D abbreviated twice.

Reply: Sorry for the mistakes. We have revised it in the updated manuscript.

References:

1. Wang Z., *et al.* Solving oxygen embrittlement of refractory high-entropy alloy via grain boundary engineering. *Mater.Today* **54**, 83-89 (2022).
2. Meisenkothen F., Steel E. B., Prosa T. J., Henry K. T., Prakash Kolli R. Effects of detector dead-time on quantitative analyses involving boron and multi-hit detection events in atom probe tomography. *Ultramicroscopy* **159 Pt 1**, 101-111 (2015).
3. Yu C., *et al.* Plastic and low-cost axial zero thermal expansion alloy by a natural dual-phase composite. *Nat. Commun.* **12**, 4701 (2021).
4. Li W., *et al.* A seawater-corrosion-resistant and isotropic zero thermal expansion (Zr,Ta)(Fe,Co)₂ Alloy. *Adv. Mater.* **34**, e2109592 (2022).

Reviewer #2 (Remarks to the Author):

Most concerns have been addressed. However, one major issue regarding boron segregation at the interface remains unclear, which is critical to the paper's main conclusion.

Reply: We sincerely thank the reviewer for the insightful comment and for recognizing our efforts in addressing the previous concerns. We agree that the clarification of boron segregation at the interface is essential to support the main conclusion.

To address this point, accordingly, we have re-analyzed and reorganized the relevant discussion.

***Comment 1:** In Figures R3 and R4, the authors continue to claim B segregation at the interface based on the proximity histogram. However, such features may also arise from artefacts due to field differences and the way the contour map was constructed. This get worse in the proximity histogram. Therefore, it is essential to present the atomic-density-distribution profile and/or histogram. This profile can be generated directly in IVAS software or extracted as an .xls file, this is one of the ways to check there was a field change at the interface.*

Reply: Thanks for your professional comment. After further discussion with our collaborators, we acknowledge that drawing quantitative conclusions based on a proximity histogram is not sufficiently rigorous, as artifacts introduced by field differences are difficult to eliminate in APT analyses. In the revised manuscript, we have reorganized the corresponding statement and made the following changes:

(i) We have removed the description of proximity histogram data determining boron atoms interface segregation, including the 2% change in value at the interface.

(ii) And we have added a Gibbs excess calculation plot for boron atoms to

Figure 4h, and used the results of the Gibbs excess calculation as the main support. To make it more rigorous, we have revised the terminology from ‘**B segregation**’ to the more appropriate ‘**B enrichment**’ throughout the entire manuscript. The specific details are: “Additionally, EPMA and atom probe tomography (3D-APT) were combined to investigate compositional fluctuations at the phase interface (Fig. 2d-f) ^{1, 2}. Boron atoms were **detected at the phase interface**, as shown in Fig. 2g-h. And the comparison of Gibbs interfacial excess, $\Gamma_B = 4.17 \pm 0.26 \text{ nm}^{-2}$ and $\Gamma_{Cr} = -0.93 \pm 0.26 \text{ nm}^{-2}$, further confirms **the enrichment of boron atoms at the interface** (Supplementary Fig. 14 and Supplementary Section 6) ^{3, 4}, suggesting its potential contribution to interfacial strengthening ⁵⁻⁷.”

Fig. R1 Microstructure of the dual-phase alloys. **a**, Rietveld refinement of the SXRD profile for the CFGB₁ alloy. **b**, EPMA image of the CFGB₁ alloy, showing the BCC matrix (gray grains) and Cr₂B precipitates (black grains). **c**,

HAADF-STEM of the Cr₂B phase along the [1 $\bar{1}$ 0] zone axis. **d**, 3D APT results determined by TEM. **e**, Cr atom maps reconstructed using 3D-APT, highlighting the nanoprobe phase interface with the BCC matrix (bottom) and the Cr₂B phase (top). **f**, Detailed view from the selected yellow box in **e**, illustrating the distributions of Fe, Ge, and B atoms. **g**, Two-dimensional concentration contour of B atom distribution. **h**, Proximity histogram for the 4-at.% B iso-concentration surfaces by using the complete probe, showing the distribution of Cr, Fe, Ge, and B atoms. Enlarged the calculation of Gibbs interfacial excess of boron. **i–j**, Segregation energy and strengthening energy at the phase boundaries with boron atoms at various sites, including an inset of the crystal structure model. **k**, Charge density distribution at the phase boundary when boron atoms occupy interstitial sites.

(iii) We have added a detailed discussion in “**Supplementary Section 6 / Confirmation of boron atom enrichment behavior at the interface**” in the supplementary information. The specific details are:

“Based on combined SXR, EPMA, and EBSD analyses, the phase boundaries between the BCC matrix and Cr₂B phases are characterized as follows: (i) the two phases exhibit markedly different crystal structures with no coherent or semi-coherent relationship; (ii) the two phases exhibit random crystallographic orientations with no defined interface relationship; (iii) Cr₂B predominantly precipitates at grain boundaries, accounting for over 90 % of all interfacial connections. This raises a critical question: given the large differences in orientation and crystal structure between the BCC/Cr₂B phase interface, why does the sample exhibit improved plasticity instead of failing due to interfacial brittleness during deformation? Although stacking fault deformation within the Cr₂B phase can release some interfacial stress, a stable interfacial structure is indispensable.

Therefore, we employed atom probe tomography (APT) to analyze the phase interface structure between BCC and Cr₂B (Fig. 2d-h). A region of interest (yellow cube) was selected at the interface to facilitate detailed analysis of the elemental distribution (Fig. 2e and f). The spatial distribution map of boron atoms shows their enrichment at the phase boundary (Fig. 2g, red spheres represent B atoms). Notably, the phase boundary (dashed line) was objectively determined based on elemental distribution mapping.

To quantify the boron enrichment at the interface more objectively, we further

analyzed the Gibbs interfacial excess of the interface. Specifically, we selected the boron 7-at.% iso-concentration surface to calculate the proximity histograms (Supplementary Fig. 14a and b), which corresponds to a more appropriate interface position (Supplementary Fig. 14c). Indeed, B is enriched between the BCC/Cr₂B boundary of $4.17 \pm 0.26 \text{ nm}^{-2}$, while Cr is considerably depleted between $-0.93 \pm 0.26 \text{ nm}^{-2}$. Fe and Ge exhibit no enrichment or depletion of $-0.04 \pm 0.01 \text{ nm}^{-2}$ and $0.15 \pm 0.00 \text{ nm}^{-2}$, as shown in Supplementary Fig. 14d-g. This further clarifies the behavior of boron atoms enriching at grain boundaries.”

Comments 2: From the contour map alone, the interface does not appear to show distinct B segregation. Instead, it exhibits a gradual increase in B concentration and then reaches Cr₂B conc., rather than sharp interfacial segregation at Cr₂B near interface. Moreover, the unit of the color pixels should be expressed as at.%/nm², not simply at.%. And please enlarge the contour map so that the B is concentrated at the near interface/Cr₂B region.

Reply: (i) For this point, “From the contour map alone, the interface does not appear to show distinct B segregation. Instead, it exhibits a gradual increase in B concentration and then reaches Cr₂B conc., rather than sharp interfacial segregation at Cr₂B near the interface. Also, in the current contour map, a red zone is still visible in the region away from the interface.”

As discussed in the revised manuscript, the positive interfacial excess of B ($\Gamma_B = 4.17 \pm 0.9 \text{ nm}^{-2}$) quantitatively confirms the enrichment of B at the interface, consistent with the description above. As a result, we have revised the terminology from ‘**B segregation**’ to the more appropriate ‘**B enrichment**’ throughout the entire manuscript.

In addition, we would like to clarify that the present system differs fundamentally from conventional grain boundaries such as BCC/BCC or FCC/FCC interfaces, where both sides of the boundary share similar chemical compositions and a distinct concentration bulge directly represents solute segregation.

In our case, the interface is between the BCC matrix and the Cr₂B phase, whose

bulk composition already contains a high and fixed amount of B atoms occupying specific crystallographic sites. Therefore, the boron atoms in Cr_2B can affect the determination of whether enrichment exists at the interface. More importantly, the relevant indication of enrichment is the presence of B atoms at the boundary plane between BCC and Cr_2B , not a higher overall B concentration than that of the Cr_2B phase itself.

(ii) As for the “Moreover, the unit of the color pixels should be expressed as $\text{at.}\%/\text{nm}^2$, not simply $\text{at.}\%$. And please enlarge the contour map so that the B is concentrated at the near interface/ Cr_2B region.”

In response, we have regenerated the boron atomic map, as shown in Fig. R2, and have also modified the corresponding parts of the images in the main text. In addition, we have carefully verified that the unit is indeed $\text{at.}\%$, consistent with the software output as well as with related published studies^{6, 8, 9}.

Fig. R2 Two-dimensional concentration contour of B atom distribution.

Comments 3: Please clarify the discrepancy between the proximity histogram and the Gibbs interfacial excess value. The histogram indicates an increase of ~2%, whereas the Gibbs excess is reported as 4.17 nm⁻². Are these values representing the same value?

Reply: Thanks for your comment. The observed $\Delta \sim 2\%$ in the proximity histogram reflects only a local concentration variation (in at.%). In contrast, the reported value of 4.17 nm⁻² is the interfacial excess, which is obtained by integrating the excess counts across the interface and dividing by the interface area. In our earlier manuscript version, we believed they had a linear relationship but not identical quantities.

In the revised manuscript, we have removed the intuitive description of concentration changes ($\Delta \sim 2\%$) and have instead adopted Gibbs' interfacial excess. The corresponding section has been updated as follows:

“Additionally, EPMA and atom probe tomography (3D-APT) were combined to investigate compositional fluctuations at the phase interface (Fig. 2d-f) ^{1, 2}. Boron atoms were detected at the phase interface, as shown in Fig. 2g-h. And the comparison of Gibbs interfacial excess values, $\Gamma_B = 4.17 \pm 0.26 \text{ nm}^{-2}$ and $\Gamma_{Cr} = -0.93 \pm 0.26 \text{ nm}^{-2}$, further confirms the enrichment of boron atoms at the interface (Supplementary Fig. 14 and Supplementary Section 6) ^{3, 4}, suggesting its potential contribution to interfacial strengthening ⁵⁻⁷.”

References:

1. Fletcher C., Moody M. P., Haley D. Fast modelling of field evaporation in atom probe tomography using level set methods. *Journal of Physics D: Applied Physics* **52**, (2019).
2. Larson D. J., Geiser B. P., Prosa T. J., Kelly T. F. On the use of simulated field-evaporated specimen apex shapes in atom probe tomography data reconstruction. *Microsc Microanal* **18**, 953-963 (2012).
3. Theska F., Primig S. Interfacial excess of solutes across phase boundaries using atom probe microscopy. *Ultramicroscopy* **256**, 113885 (2024).
4. Hellman O. C., Seidman D. N. Measurement of the Gibbsian interfacial excess of solute at an interface of arbitrary geometry using three-dimensional atom probe microscopy. *Mater. Sci. Eng. A* **327**, 24-28 (2002).
5. Zhou X., et al. Atomic motifs govern the decoration of grain boundaries by interstitial solutes. *Nat. Commun.* **14**, 3535 (2023).

6. Yang T., *et al.* Ultrahigh-strength and ductile superlattice alloys with nanoscale disordered interfaces. *Science* **369**, 427-432 (2020).
7. Lu L., Chen X., Huang X., Lu K. Revealing the maximum strength in nanotwinned copper. *Science* **323**, 607-610 (2009).
8. Shi P., *et al.* Bioinspired, heredity-derived hierarchical bulk multifunctional copper alloys. *Mater.Today* **71**, 22-37 (2023).
9. Jiang K., *et al.* Highly selective oxygen reduction to hydrogen peroxide on transition metal single atom coordination. *Nat. Commun.* **10**, (2019).

Reviewer #2 (Remarks to the Author):

The authors have addressed all concerns. The reviewer congratulates the authors.

Reply: We appreciate the reviewer's positive feedback and recommendation. We would like to thank the reviewer for their time and support of our work.